# Multiple Myeloma Therapy: Emerging Trends and Challenges

**DOI:** 10.3390/cancers14174082

**Published:** 2022-08-23

**Authors:** Danai Dima, Dongxu Jiang, Divya Jyoti Singh, Metis Hasipek, Haikoo S. Shah, Fauzia Ullah, Jack Khouri, Jaroslaw P. Maciejewski, Babal K. Jha

**Affiliations:** 1Department of Translational Hematology and Oncology Research, Taussig Cancer Institute, Cleveland Clinic, Cleveland, OH 44195, USA; 2Center for Immunotherapy and Precision Immuno-Oncology, Lerner Research Institute, Cleveland, OH 44195, USA; 3Department of Hematology and Medical Oncology, Taussig Cancer Institute, Cleveland Clinic, Cleveland, OH 44195, USA; 4Case Comprehensive Cancer Center, Case Western Reserve University, Cleveland, OH 44106, USA; 5Cleveland Clinic Lerner College of Medicine, Cleveland, OH 44195, USA

**Keywords:** multiple myeloma, immunotherapy, targeted therapy

## Abstract

**Simple Summary:**

Multiple myeloma (MM) is a frequent hematological malignancy characterized by the uncontrolled growth of clonal plasma cells, primarily in the bone marrow. Over the past years, novel therapies have been discovered and introduced into clinical practice that have dramatically changed the treatment landscape of MM. Despite the tremendous advances, MM remains incurable, with poor outcomes particularly in patients with relapsed/refractory disease, emphasizing the need for new therapeutic approaches. In this review we discuss the current treatment paradigm of multiple myeloma and focus on promising future approaches.

**Abstract:**

Multiple myeloma (MM) is a complex hematologic malignancy characterized by the uncontrolled proliferation of clonal plasma cells in the bone marrow that secrete large amounts of immunoglobulins and other non-functional proteins. Despite decades of progress and several landmark therapeutic advancements, MM remains incurable in most cases. Standard of care frontline therapies have limited durable efficacy, with the majority of patients eventually relapsing, either early or later. Induced drug resistance via up-modulations of signaling cascades that circumvent the effect of drugs and the emergence of genetically heterogeneous sub-clones are the major causes of the relapsed-refractory state of MM. Cytopenias from cumulative treatment toxicity and disease refractoriness limit therapeutic options, hence creating an urgent need for innovative approaches effective against highly heterogeneous myeloma cell populations. Here, we present a comprehensive overview of the current and future treatment paradigm of MM, and highlight the gaps in therapeutic translations of recent advances in targeted therapy and immunotherapy. We also discuss the therapeutic potential of emerging preclinical research in multiple myeloma.

## 1. Introduction

Multiple myeloma (MM) is the second most common hematologic malignancy, characterized by the uncontrolled growth of clonal plasma cells (PCs). These cells are hyperproliferative differentiated B-lymphocytes capable of secreting a variety of immunoglobulins (Ig) [1] with the most common being IgG, IgA, and, to a lesser extent, IgD. The abnormal PCs typically grow in the bone marrow with only a small fraction of patients presenting with extramedullary disease at the time of diagnosis or developing extramedullary disease later in the disease course [2,3]. The excessive monoclonal Ig secretion from the clonal PCs leads to organ damage with the most frequent clinical symptomatology including anemia, renal failure, hypercalcemia, and lytic bone lesions [4].

Over the past few decades, rigorous pre-clinical and clinical research has led to the discovery of novel therapies that have dramatically changed the treatment landscape of MM in the frontline as well as the relapsed/refractory setting (Figure 1). The introduction of these agents has translated into prolonged progression-free periods as well as overall survival with significantly less toxicity and improved quality of life [5,6]. However, despite these tremendous advances, MM remains largely incurable and very heterogeneous, with poor outcomes especially among patients who are resistant to multiple drug classes, emphasizing the need for a better understanding of its underlying biology and the development of more effective therapeutic strategies. We will discuss herein the current treatment paradigm of multiple myeloma and focus on promising future approaches.

## 2. Evolution and Molecular Basis of Multiple Myeloma

Recently, there has been an increasing interest in unraveling the underlying genomics of MM in an effort to better understand the clinical heterogeneity of this complex disease, characterize its precursor states, and elucidate the significant cytogenetic and molecular events leading to active myeloma. Understanding the molecular basis of myeloma is also believed to improve risk stratification, prognostication, and prediction of response to available therapies, altogether advancing precision medicine.

At present, it is well known that MM is a progressive disorder with distinct developmental stages. Monoclonal gammopathy of undetermined significance (MGUS), smoldering multiple myeloma (SMM) and active MM form the disease continuum characterized by progressive clonal evolution [7,8]. The gradual accumulation of cytogenetic abnormalities with translocations (such as IgH translocation) and/or gains and/or losses of chromosomes (such as loss of 1 p, 17 p, monosomies 13, 14, 17 or 1 q gain), as well as genetic mutations and epigenetic alterations in PCs over time, lead to disease progression ultimately resulting in the malignant phenotype [9,10,11,12].

Within the expanding tumor, there are genetically distinct subclones generating fluctuating intertumoral heterogeneity with particular subclones gaining selective advantage and dominating. Comparison of whole-genome sequencing data (WGS) of patients at precursor stages and with active MM showed that the sub-clonal heterogeneity is present in the early stages of MGUS and persists throughout the disease course, suggesting the presence of a branching clonal evolution pattern [13,14]. Microenvironment may also play a role in the transition from precursor disease to symptomatic myeloma [15].

MM is nearly always proceeded by MGUS; however, only ~10% of patients with MM have a prior MGUS diagnosis. This may be related to the fact that MGUS is asymptomatic and most cases are incidentally found on routine checks. The annual risk of MGUS progression to MM is around 1%; however, for high-risk MGUS, the risk almost doubles at 2% [16]. Recent findings suggest that the likelihood of conversion to a higher risk MGUS state (defined by an increase in serum biomarkers, Figure 2) can change over time, supporting continued follow up and risk assessment in MGUS patients [17].

Smoldering myeloma is a very heterogenous intermediate stage, between MGUS and MM, with variable risk for progression. Many cases are shown to have a low mutational burden and called MGUS-like; however, a high mutational burden, similar to symptomatic MM, has also been observed [7]. A phase 2 trial of carfilzomib, lenalidomide, and dexamethasone in high-risk SMM performed whole-exome sequencing (WES) of 18 SMM patients. When compared to newly diagnosed MM (NDMM), SMM patients had a significantly lower frequency of mutations in significant myeloma and NF-κB pathway genes [18].

To explore the genomic changes at the level of active disease, a study from Chapman, et al. evaluated 38 patients with NDMM, using either WGS (23 patients) or WES (16 patients). They found frequent mutations in genes known to be involved in myelomagenesis (*NRAS*, *KRAS*, *TP53*, *CCND1*), protein translation (*DIS3*, *FAM46C*), histone methylation, and the *NF-κB* pathway [19]. A larger study from the same group using the next-generation sequencing (NGS) technique for WGS (177 patients) or WES (26 patients), adding valuable information regarding the clonal architecture of MM and reporting the presence of different sub-clonal populations frequently harboring multiple mutations inside the same pathway (e.g., *KRAS*, *NRAS*) within the same patient [20].

WES of 463 patient with NDMM of the Myeloma XI trial also identified significant alterations in 15 genes including *KRAS*, *NRAS*, *TP53*, *DIS3*, and *IRF4*. Mutations in the *RAS* and *NF-κB* pathways were common at 43% and 17% respectively; however, they were not associated with the prognosis [10]. On the contrary, mutations in *CCND1* and the DNA repair pathway (*TP53*, *ATM*, *ATR*, and *ZNFHX4*), chromosome 1q amplification, and *MYC* translocations were associated with inferior progression-free and overall survival outcomes [10].

NGS for WES had also been utilized in patients with advanced disease, confirming the high variability of genomic architecture in MM and additionally describing a significant rate of inactivating mutations in *SP140* and *LTB* tumor suppressor genes. Increased risk of relapse was noted with *SP140* mutations [21]. It was further shown that heterogeneity of the sub-clonal structure and mutational reserve of MM is generated by at least two mutational processes, which can independently change/grow over time, demonstrating distinct branching clonal evolution. Higher numbers of mutations have been associated worse survival outcomes, regardless of how mutations are generated or distributed across subclones [21].

## 3. Factors Influencing Treatment Strategy and Current Challenges

Several patient-related, disease-related, and treatment-related factors should be considered when selecting therapy for MM. The most important patient-related factors include age, frailty, and performance status. Disease-related parameters include the nature of the disease, for example risk status, and extent of organ damage. Treatment-related factors refer to drug availability and drug adverse events; in the case of relapsed/refractory disease the number and type of prior treatments, as well as depth and duration of previous responses, should also be taken into consideration.

Risk stratification at diagnosis is important for treatment selection. There are several scoring systems that risk stratify myeloma and its predecessor states (Figure 2). The simplest and most commonly applied system for MM is the International Staging System (ISS), which uses two parameters: the serum β-2 microglobulin and serum albumin at diagnosis [22]. The revised version referred to as R-ISS additionally incorporates high-risk chromosomal abnormalities (del (17 p), t (4;14), and t (4;16)) and serum lactate dehydrogenase [23]. Recent data for the Myeloma Genome Project that utilized NGS for WGS and WES reported that the biallelic inactivation of the TP53 gene or amplification (≥4 copies) of *CKS1B* (1q chromosome) in patients with ISS III were associated with poor prognosis; however, these molecular tests are not routinely done [24].

In the era of NGS utilization, several gene expression profiles (GEP) have attempted to risk stratify patients with NDMM. However, so far only the EMC-92 and UAMS GEP70 have been validated for their prognostic utility and developed into clinical tests [25,26]. A recent study by Shah et al., applied the EMC-92 GEP to the 329 NDMM transplant-eligible patients of the Myeloma XI trial and found that the EMC-92 was able to precisely identify patients with high-risk MM [27]. Moreover, they showed an independent association of EMC-92 and high-risk cytogenetic aberrations with clinical outcomes. Despite this encouraging ability for accurate risk stratification, molecular signatures are still in the embryonic stages with regards to application to clinical practice.

With respect to regimen selection, if patients are fit enough, three-drug regimens are always preferred over two-drug regimens, as they have consistently shown better responses and survival outcomes. However, frail patients may only be able to tolerate duplet therapy. Despite implementing multimodal approaches to treat MM, the major challenge remains that the vast majority of patient eventually relapse and become refractory to multiple drug classes. Additionally, patients require continuous treatment throughout the disease course, which can negatively affect their quality of life due to potential therapy-related side effects. Some patients, due to significant comorbidities, may not be candidates for specific systemic agents or autologous transplantation, which can significantly limit the suitable pool of modalities that can be utilized against their disease. Moreover, despite recent introduction of several novel agents into routine practice, there are no reliable markers that can successfully predict the response to specific drug classes, restricting our ability to select a more personalized therapeutic strategy.

## 4. Traditional Therapies

### 4.1. Alkylating Agents

Historically, alkylating agents have been used for the treatment of MM. Most alkylators work similarly by directly damaging DNA. In detail, they cause crosslinking and subsequent DNA strand breaks. This leads to abnormal base pairing, inhibiting cell division and resulting in apoptosis [28].

Melphalan was the first alkylator and first drug ever found to be effective against MM. Melphalan along with prednisone demonstrated a significant benefit and was eventually established as the standard of care in 1969 [29]. Given the introduction of novel agents over the past 20 years, melphalan-based regimens are rarely used in the US now, with two exceptions only: (1) high-dose melphalan is still the preferred conditioning regimen prior to autologous hematopoetic cell transplant (AHCT) and (2) it is used in combination with bortezomib, dexamethasone, cisplatin, doxorubicin, cyclophosphamide, and etoposide (VTD-PACE) for very aggressive disease.

Cyclophosphamide is another popular alkylator, currently used in combination with other standard-of-care agents. Cyclophosphamide has been shown to additionally have a significant immunomodulatory effect by activating NK cells, T helper cells, and macrophages [29]. Melflufen is a newer peptide–drug conjugate that targets aminopeptidases and rapidly releases the alkylator payload into the MM cells. [30] This irreversibly damages DNA and results in cell death via a p53-independent mechanism. It was recently granted accelerated approval by the FDA; however, shortly thereafter, it was withdrawn due to inferior survival compared to standard of care. Further trials are required to explore its potential.

### 4.2. Immunomodulatory Drugs

Immunomodulatory drugs (IMiD) have been an important component of the backbone of therapy for MM. The first-generation IMiD Thalidomide was introduced into clinical practice in the late 1990’s, followed by approval of the next-generation IMiD Lenalidomide and Pomalidomide. These were observed to be more effective with a higher therapeutic index, justifying their current widespread use in routine clinical practice [31].

IMiD hold a pleiotropic mechanism of action, including direct cytotoxicity via cell cycle arrest, induction of apoptosis as well as inhibition of cell adhesion molecules, and expression of early growth genes [31]. The cytotoxicity of IMiD is mediated by their binding to cereblon, an adapter protein linked to E3 ubiquitin protein ligase complex CRL4. This interaction induces ubiquitination and degradation of the transcription factors Ikaros and Aiolos, which are important for lymphocyte development and proliferation, ultimately resulting in MM cell apoptosis [32,33].

In addition, IMiD have immunostimulatory activity leading to increased CD4+ and CD8+ T-cell co-stimulation and secretion of IL-2 and IFN-γ, which can lead to activation of the Natural Killer (NK) cell compartment and antibody-dependent cellular cytotoxicity (ADCC) [34,35,36,37]. NK cell activation by IMiD has also been shown to occur via activation of Zap-70 and degradation of both Ikaros and Aiolos [38]. Of note, Ikaros and Aiolos suppress IL2 production in immune cells, thus IMiD increase the production of IL2 leading to T and NK cell proliferation and suppression of regulatory T cell (Tregs) function [38,39]. IMiD can also boost the immune response by enhancing antigen presentation and increasing the activity of dendritic cells (DCs) [31,40]. They also interfere with the tumor micro-environment by disrupting angiogenesis, adhesion molecules, and the stromal-MM cell interaction, altering cytokine and growth factor secretion and inhibiting osteoclastogenic proliferation [31].

Multiple preclinical and clinical trials have confirmed the efficacy of IMiD. They are currently approved as monotherapy or in several combinations with one or two or three other anti-myeloma agents in the frontline, maintenance, and the relapsed settings (Table 1).

A next generation of immunomodulators, called cereblon E3 ligase immunomodulators (CELMoDs), is presently under clinical investigation. CELMoDs and IMiD have a similar mechanism of action but CELMoDs differ in that they bind to cereblon more avidly, thus lowering its cellular concentration more potently than IMiD [41]. This leads to a much deeper and durable depletion of Ikaros and Aiolos, which are overexpressed in MM, promoting tumor growth and proliferation.

The CELMoD Iberdomide (also known as CC-220) has shown enhanced tumoricidal and immune stimulatory effects in preclinical studies [42]. The first phase 1/2, open-label, dose-escalation study of Iberdomide in combination with dexamethasone in heavily pretreated MM patients (mostly IMiD resistant) demonstrated an overall response rate of 32% and good tolerability [43]. Patients who were refractory to other IMiD had a similar ORR of 35%. Iberdomide was then combined with daratumumab, or bortezomib or carfilzomib (NCT02773030), with good responses and a favorable safety profile, with very few grade 3–4 adverse events. ORR were 46%, 56%, and 50% for the three Iberdomide-based combinations, respectively [44]. It appears that Iberdomide might be able to overcome resistance to both lenalidomide and pomalidomide, and this is awaited to be confirmed in future phase 3 randomized trials.

Another novel CELMoD that works similarly to Iberdomide is CC-92480, which has demonstrated potent synergy and safety when combined with dexamethasone in the phase 1 CC-92480 MM-01 trial (NCT03374085) that included heavily pretreated patients with relapsed/refractory MM (RRMM) [45,46]. The positive outcomes resulted in expansion of this study to CC-92480-MM-002 (NCT03989414) that currently evaluates the combination of CC-92480 with other agents including proteasome inhibitors and the anti-CD38 monoclonal antibody. Preliminary results are encouraging. Avadomide (CC-122), another cereblon-modulating agent with potent biological activities, was tested in a phase 1 trial and demonstrated acceptable safety and favorable pharmacokinetics (NCT01421524) [47]. CFT7455, a next-generation Ikaros/Aiolos degrader, is currently being tested in a phase 1 trial with dose escalation and dose expansion phases (NCT04756726) [48].

### 4.3. Proteasome Inhibitors

Proteasome Inhibitors (PI) have been an instrumental therapeutic class in MM. They primarily act by disrupting the natural degradation of intracellular proteins via inhibition of the proteasome complex inducing direct apoptosis. PI also downregulate the NF-κB signaling pathway and block the activation of anti-apoptotic genes, leading to cell death [49,50]. Additionally, PI have other properties including suppression of adhesion molecules and angiogenesis [51]. There is recent evidence that PI can have an immune stimulatory effect including activation of NK cells and DCs, which ultimately results in immune-mediated destruction of the malignant PCs [52,53,54].

At present, there are three PI approved for the treatment of MM—bortezomib (V), carfilzomib (K), and ixazomib [55]. Bortezomib, a first-generation PI, has a slow reversible effect on the proteasome, whereas carfilzomib, a second-generation PI, binds the proteasome irreversibly and more selectively [55,56]. Additionally, V inhibits both chymotrypsin-like and caspase-like activities of the proteasome, whereas K is more selective for the chymotrypsin-like active site within the proteasome [56]. Therefore, K holds greater specificity and potency and a different safety profile compared to V. Ixazomib is the only orally bioavailable PI, with properties similar to V, but with a more begnin neutotoxicity adverse event profile [57,58]. PI are typically used in doublet, triplet, or quadruplet combinations with other drug classes such as IMiD and monoclonal antibodies for newly diagnosed as well as RRMM (Table 1).

Other, newer PI, still under investigation in the preclinical or phase 1/2 level, are marizomib, oprozomib, and delanzomib. Marizomib is a potent irreversible inhibitor of all three core enzyme activities of the 20 S proteasome [59]. It is currently studied as a single agent or in combination with pomalidomide and dexamethasone in previously treated RRMM patients. Oprozomib is an orally bioavailable PI that primarily inhibits the chymotrypsin-like activity of the proteasome and works similarly to carfilzomib [60,61]. It is being studied as monotherapy or along with IMiD, steroids, or alkylating agents. Two other oral PI, LC53-0110 and LC53-0151, have only been studied in mice xenografts, where they have demonstrated remarkable potency [62]; however, they have not been tested in humans yet.

## 5. Immunotherapy

### 5.1. Immune System Dysreguation

A hallmark of the underlying biology of MM is the immune system dysfunction, which is caused by various mechanisms and is believed to play a central role in the pathogenesis of the disease by promoting clonal cell proliferation via immune escape and contributing to drug resistance [63,64]. Loss of tumor antigenicity via impaired expression or alterations of tumor antigens on the surface of MM cells and upregulation of inhibitory surface ligands can lead to tumor escape from immune surveillance, along with defects in antigen processing/presentation [65,66,67,68,69,70,71,72]. This is supported by the robust T-cell response to MM antigens in bone marrow samples of patients with MGUS but absence of this phenomenon in the bone marrow of patients with active MM, despite similar clonal PCs populations [73,74].

The dysregulation further involves the tumor microenvironment, including alterations in the T and NK compartments, with upregulation of inhibitory molecules/ligands, resulting in an immunosuppressive milieu [75,76,77,78,79,80]. The increased recruitment of immunosuppressive cells such as Tregs, regulatory B cells (Bregs), tumor-associated macrophages (TAMs), and myeloid-derived suppressor cells (MDSCs), along with the simultaneous reduction in cytotoxic T lymphocyte (CTLs) and defective function of antigen-presenting DCs, leads to decreased humoral and cytotoxic immunity [76,80,81,82,83]. Several novel agents that have been developed over the past decade are subsumed under the umbrella of immunotherapy and target different aspects of the immune system to eradicate MM cells. These agents include monoclonal antibodies, immune checkpoint inhibitors, bispecific antibodies, genetically engineered immune cells, and peptide vaccines.

### 5.2. Naked Monocloal Antibodies

Naked monoclonal antibodies (mAb) target antigens primarily expressed on the surface of PCs and lead to cell death via several different mechanisms. Currently, there are multiple antigens studied as potential targets, with the most important being CD38 and signaling-lymphocyte-activating molecule family-7 (SLAMF7), against which mAb have been developed are broadly used in clinical practice.

#### 5.2.1. Anti-CD38 mAb

Daratumumab is a humanized mAb that targets a unique epitope on the CD38 transmembrane glycoprotein, predominately located at the surface of PCs [84]. CD38 is a multifunctional ectoenzyme that modulates NAD^+^ degradation and plays a critical role in the synthesis of cyclic-ADP-ribose, properties that contribute to cell survival via intracellular calcium mobilization and homeostasis [84,85,86,87,88].

The binding of daratumumab to its target induces cell death through various immune effector mechanisms. These mechanisms include complement-dependent cytotoxicity (CDC), antibody-dependent cellular cytotoxicity (ADCC), antibody-dependent cellular phagocytosis, and apoptosis via crosslinking [89,90,91]. Moreover, inhibiting the ectoenzymatic function of CD38 can lead to the direct apoptosis of neoplastic cells [88]. Another mechanism of daratumumab action is linked to its immunomodulatory activity, due to the presence of CD38 on various immune cells such as Tregs, Bregs, and MDSC [92].

There is significant clinical evidence for the high efficacy of daratumumab as a single agent or in combination with other drug classes [93,94,95,96] (Table 1). Pre-clinical studies have demonstrated its synergistic action with IMiD, which upregulate the expression of CD38 on PCs and therefore sensitize them to anti-CD38 mAb [97].

Isatuximab is a chimeric anti-CD38 mAb that works similarly to daratumumab [89]. Isatuximab binds to a different epitope of the CD38 surface receptor, alleviating the immunosuppressive microenvironment of the bone marrow niche in MM [98,99]. It is currently approved in combination with the duplets of pomalidomide–dexamethasone and carfilzomib–dexamethasone for RRMM (Table 1), with ongoing clinical trials testing various other combinations in the relapsed and frontline settings [100,101].

#### 5.2.2. Anti-SLAMF7 mAb

Signaling-lymphocyte-activating molecule family-7 (SLAMF7) is a receptor that is highly expressed on the surface of PCs but also other leukocytes, especially NK cells [102,103,104,105]. It has immunomodulatory properties and mediates the interaction of MM cells among each other and also with tumor microenvironment stroma, activating several downstream pathways that promote malignant PCs proliferation. [106,107].

Elotuzumab is an anti-SLAM7 humanized mAb whose Fc portion binds to the CD16 on the surface of NK cells, constituting a bridge between MM and NK cells [108,109]. This attachment leads to the activation of NK cells, tagging MM cells for destruction in an ADCC-dependent manner [106]. There is preclinical evidence that augmenting elotuzumab NK cell-mediated ADCC can enhance its anti-tumor effect. This can be achieved by the blockade of the killer immunoglobulin-like checkpoint receptors (KIR) with the experimental mAb lirilumab, or by activating the CD137 receptor with the mAb Urelumab. However, these combinations are yet to be studied in the clinical context [110,111]. Elotuzumab can also directly activate NK cells by binding to SLAMF7 on their surface, hence facilitating the downstream activation of EAT2, leading to enhanced phosphorylation of ERK [112]. Given that the activity of elotuzumab is strongly dependent on the activity of NK cells, dysfunction in the NK compartment can lead to decreased efficacy. Moreover, elotuzumab can interfere with the tumor microenvironment by disrupting the adhesion of MM cells to the bone marrow stromal cells [107].

Initial experimental studies demonstrated that elotuzumab is more effective when combined with traditional antimyeloma agents such as lenalidomide and bortezomib [113,114]. This was further confirmed in clinical trials which found elotuzumab to have modest activity as monotherapy [115], and better activity when combined with lenalidomide or pomalidomide [116,117]. Elotuzumab is currently approved with pomalidomide/dexamethasone or lenalidomide/dexamethasone for relapsed/refractory disease (Table 1). Elotuzumab has also been studied in combination with bortezomib (NCT01478048) with encouraging results. A phase 3 trial exploring the addition of elotuzumab to the triplet carfilzomib/lenalidomide/dexamethasone in the upfront and post-transplant maintenance setting is ongoing (NCT03948035).

### 5.3. Immune Checkpoint Inhibitors

Immune checkpoints are inhibitory receptors on the surface of T cells that mediate immune tolerance to self-antigens, by suppressing the T cell compartment when activated during the antigen presenting process [118,119]. There is evidence that the cytotoxic tumor lymphocyte antigen 4 (CTLA-4) and programmed cell death-1 (PD-1) immune checkpoints are highly expressed on the surface of T, B, and NK cells in the bone marrow of patients with MM [120,121]. The binding of these receptors to their ligands on antigen presenting cells (APC) and/or tumor cells leads to suppression of cytotoxic T cells and upregulation of Tregs, thus inhibiting the immune response, favoring cancer cell growth via immune escape [119,120,122]. Experiments have shown that PD-L1, the major ligand of the PD-1 receptor, can be upregulated on malignant PCs [122]. MM cells with high PD-L1 expression appear to be more proliferative and resistant to therapy, indicating increased aggressiveness [123]. T cell immunoreceptor with Ig and ITIM domains (TIGIT) and lymphocyte activation gene-3 (LAG-3 or CD223) are other immune checkpoints on the surface of T cells involved in T cell regulation by activating Tregs and inhibiting cytotoxic T cells [124,125,126,127]. Increased expression of LAG3 on T cells in the bone marrows of MM patients is associated with sustained T cell stimulation leading to T cell exhaustion, which can potentially contribute to immune escape [128].

Immune checkpoint inhibitors (ICI) are a distinct category of “naked” mAb targeting molecules that constitute immune checkpoints. The ICI mainly evaluated in MM are the anti-PD-L1 mAb, which block the binding of PD-L1 to its PD-1 receptor. PD-1/PD-L1 blockade alone was not efficacious in phase 1 studies [128,129]; however, combination approaches with different drug classes such as IMiD and anti-CD38 mAb appeared promising. In preclinical studies, IMiD reduced the expression of PD-1 receptors on T cell surfaces and also down-regulated PD-L1 on MM cells, supporting a potential synergetic effect with PD-L1 inhibitors [130]. In vivo studies have also shown that long exposure to PD-1 blockade enhances the anti-CD38 ADCC, suggesting a potential clinical benefit to combining anti-PD-L1 ICI with anti-CD-38 mAb [131].

Clinically, several studies have evaluated PD-1 blockade using pembrolizumab or nivolumab with IMiD such as lenalidomide and pomalidomide; however, they failed to demonstrate improvement in disease response. Interestingly, a combination of pembrolizumab with lenalidomide was associated with high rates of toxicity and increased risk of death; thus, the FDA put a hold on studies investigating combinations of anti-PD-L1 ICI with IMiD [132,133,134]. Anti-PD-L1 inhibitors have also been used in combination with daratumumab without safety warnings but no clinical benefit so far [135,136,137,138]. Nivolumab is currently being tested in combination with carfilzomib and pelareorep in a phase 1 trial (NCT03605719). More recent phase 1 and 2 trials are examining the efficacy and safety of anti-LAG 3 (BMS-986207) and anti-TIGIT (BMS-986207 or COM902) ICI alone or combined with other agents (NCT04354246, NCT04150965). To date, the use of ICI alone or in combination with traditional anti-myeloma agents has not proven efficacious in the clinical setting and is currently not recommended.

### 5.4. Antibody Drug Conjugates

A novel type of therapy that has recently been investigated in the clinical setting is the antibody drug conjugates (ADC). ADC are mAb against a specific tumor target on the surface of malignant cells that carry a small cytotoxic agent (payload), such as microtubule inhibitors and agents damaging DNA, utilizing a cleavable or non-cleavable linker [139,140]. When it reaches its target, the ADC is internalized with eventual release of the payload into the cytoplasm of malignant PCs, leading to cell death [140] (Figure 3). Cleavable linkers are degradated by enzymes in the cytoplasm of the malignant cells, whereas non-cleavable linkers require processing and degradation of the mAb complex into the lysosomes in order to release the toxic payload [140]. The target of ADC should ideally be a molecule highly expressed on the surface of malignant PCs with very low or no expression on other cell types, including hematopoeitic cells, to avoid systemic toxicity [141]. ADC can also exert their effects via ADCC, ADCP, or CDC [142,143].

#### 5.4.1. B Cell Maturation Antigen (BCMA)

The most well-studied ADC target to date is BCMA, which is a member of the tumor necrosis factor receptor family exclusively expressed on plasma cells and late mature B cells [144]. Activation of BCMA by its ligands, B cell activating factor (BAFF) and a proliferation-inducing ligand (APRIL), is required for the survival of PCs [145,146]. At present, there are several different ADCs under development that target the BCMA carrying different types of linkers and payloads.

The anti-BCMA ADC belantamab mafodotin (GSK2857916) is a humanized IgG1 anti-BCMA mAb carrying a non-cleavable linker to the tubulin polymerization inhibitor monomethyl aurastatin-F (mcMMAF) [143]. Apart from the classic mechanism of cell death after the endoplasmic release of mcMMAF, belantamab mafodotin can also induce cell lysis via effector-cell-mediated ADCC, with the defucosylated Fc region of the mAb helping to facilitate this interaction [143,147]. Malignant cell lysis releases tumor antigens, locally triggering a robust immune response against myeloma cells. [146,147]. The clinical safety and efficacy of belantamab mafodotin monotherapy in heavily pretreated MM patients were confirmed in the DREAMM-1 and DREAMM-2 clinical trials [148,149,150]. At present, belantamab mafodotin is being studied in combination with other anti-myeloma agents in phase 2 and 3 clinical trials (NCT03715478, NCT03544281, NCT04246047 and NCT04484623). To date, it is the only ADC approved by the FDA, as a single agent, for the treatment of MM patients who have received at least four prior therapies including an anti-CD38 mAb, a PI, and an IMiD [149,151,152].

Other anti-BCMA ADC include AMG224, MEDI2228, CC-99712, and HDP-101. AMG224 is a mAb conjugated with antitubulin maytainsinoid using a noncleavable linker, currently studied as monotherapy in a phase 1 trial in heavily pretreated MM patients (NCT02561962) [153,154]. MEDI2228 is another BCMA-targeted ADC conjugated with a DNA cross-linking pyrrolobenzodiazepine using a protease cleavable linker [155]. After binding to BCMA, the complex is internalized and cleaved in the lysosomes, releasing the toxic payload, which causes DNA damage and apoptosis [155]. In preclinical studies, it demonstrated synergism with bortezomib [156,157]. A phase 1 study assessed its safety as a single agent in the RRMM setting, reporting encouraging clinical efficacy and a manageable safety profile (NCT03489525) [158]. The newer anti-BCMA ADCs, CC-99712 and HDP-101, are also in early phase studies (NCT04036461, NCT04879043).

#### 5.4.2. Other ADC Targets

CD38 is a glycoprotein on the membrane of PCs targeted by the “naked” mAb daratumumab and isatuximab, with robust clinical efficacy [85]. MT-0169 or TAK-169 is an investigational ADC that similarly targets the CD38 receptor; however, it is conjugated with a ribosome-inactivating Shiga-like engineered toxin called (SLTA) [159]. In vivo pre-clinical data have been encouraging, leading to a phase 1 (NCT04017130) study. TAK-573 is another anti-CD38 IgG4 ADC genetically fused to two attenuated interferon alpha-2b (IFNα2b) molecules. Notably, this agent binds to a location of the CD38 receptor that is different from the binding sites of daratumumab and isatuximab [160]. Given promising outcomes in xenograft models, there is an ongoing a phase 1/2 trial testing TAK-573 in combination with dexamethasone (NCT03215030).

CD46 is a transmembrane protein that is highly expressed in MM cells that is involved in complement inhibition [161]. An anti-CD46 ADC is fused with saporin and subsequently to monomethylauristatin F (MMAF) [161]. Preclinical data have demonstrated anti-myeloma activity, although testing in humans has not been initiated yet. FOR46, another anti-CD46 ADC with an undisclosed payload, is currently being tested in a phase 1 trial for RRMM (NCT03650491).

CD74 is a glycoprotein found on the surface of the majority of PCs and is involved in the antigen presentation process by the major histocompatibility complex class II [162,163]. STRO-001 is a human anti-CD74 IgG fused with a potent maytansinoid warhead to two specific sites [164]. This agent is being studied in an ongoing phase 1 trial (NCT03424603) with encouraging preliminary results [165].

### 5.5. Bispecific Antibodies

Bispecific antibodies (bsAbs) are mAb designed to bind to a target on the surface of the malignant myeloma cells and effector cells (T or NK cells), creating an immunologic bridge leading to the destruction of the tumor cell by the activated effector cell [166] (Figure 3). There are several bsAbs currently being tested in the preclinical and clinical settings. The most popular antigenic targets on PCs include BCMA, CD38, GPRC5D, and FcRH5 [167]. GPRC5D is a G-protein–coupled receptor with unclear function that is highly expressed on the surface of myeloma cells [168]. FcRH5 belongs to the immunoglobulin superfamily and is located only on the surface of B cells with increasing expression on myeloma cells [169,170]. At present, all bsAbs in clinical trials target the CD3 on the surface of T cells [167]. However, in a preclinical level, bispecific NK-cell engagers are also under investigation with good anti-myeloma activity [171,172,173].

#### 5.5.1. Bispecific T Cell Engagers

There are several bispecific antibodies targeting BCMA and CD3 on T cells (BCMAxCD3) that induce T cell activation against the tumor and lead to lysis of MM cells [174]. The main ones currently being studied in the clinical setting include: AMG420, AMG701, CC-93269, elranatamab (PF-06863135), REGN5458, REGN5459, teclistamab (JNJ-64007957), and TNB-383B [175,176,177,178] (Table 2). Bispecific T cell engagers targeting GPRC5D on tumor cells (GPRC5DxCD3) include talquetamab (JNJ-64407564) [179,180,181], whereas the ones targeting CD38 (CD38xCD3) and FcRH5 (FcRH5xCD3) on PCs are TNB-383B and cevostamab, respectively [182,183]. Development of trispecific antibodies or trispecific T cell engagers is a step forward for bsAbs, by adding a T cell costimulatory signal such as CD28 or dual MM targeting in an attempt to overcome the possibility of T cell anergy, which can lead to suboptimal responses [184,185]. Preclinical testing of a trispecific antibody targeting the CD38, CD3, and CD28 on T cells in mice xenografts, has demonstrated a very potent antimyeloma effect [184]. However, no trispecific antibody has entered clinical testing in humans yet. At present, several bispecific and trispecific antibodies activating the T cell compartment are under rigorous investigation and awaiting advancement to the clinic.

#### 5.5.2. Bispecific NK Cell Engagers

Bispecific NK cell engagers are bispecific antibodies that redirect the NK cell compartment against the tumor cells via binding to a receptor on the NK cell surface such as CD16A, NKp30, or NKG2d and to a molecule on the surface of the malignant PCs, such as the BCMA or SLAM7 [171,172,173]. Several agents are under development with the following combinations: BCMAxCD16a (AFM26, RO7297089), BCMAxNKp30 (CTX-8573), and SLAM7xNKG2D (CS1-NKG2D) [171,172,173,186,187]. A trispecific NK cell engager targeting the BCMA, as well as the CD20 co-stimulatory molecule and CD16a on NK cells, is also under preclinical stages of development [188]. Clinical trials of this agent class are eagerly awaited.

### 5.6. Chimeric Antigen Receptor (CAR) T Cell Therapy

Chimeric antigen receptors (CAR) are synthetic transmembrane receptors that are designed to selectively recognize specific antigens on the surface of target cells [189,190]. The extracellular antigen recognition domain typically consists of a single-chain variable fragment (scFv), whereas the intracellular activation domain is typically derived from the CD3ζ chain that subsequently induces T cell activation upon antigen binding [191,192,193] (Figure 3). First-generation CAR lacked a costimulatory domain, resulting in only moderate responses [194]. However, the next-generation CAR included co-stimulatory signaling endodomains, such as CD28, CD137 (4-1BB), or inducible T cell co-stimulator (ICOS), in an attempt to mimic the co-stimulation occurring during physiological T cell activation via TCR recognition by APC, with subsequent improvement in T cell responses [195].

The CAR T cell production starts with collection of T cells from patients and continues with the transfer of the gene encoding the CAR construct into the genome of these T cells using a viral vector [196,197]. The CAR gene is subsequently transcribed and expressed as a surface receptor [198,199]. CAR T cell manufacturing occurs ex vivo and takes 4 weeks on average [198,199]. CAR T cell therapy is typically given as a single infusion after the administration of lymphodepleting chemotherapy, which facilitates the proliferation and activity of CAR T cells [200,201].

The choice of target antigen is critical, as it needs to be uniformly expressed on malignant cells with minimal expression on other hematopoietic cells and tissues [198]. BCMA was the first antigen to be targeted in CAR T cell therapy clinical trials [145]. Idecabtagene vicleucel (Ide-cel) was the first CAR T product officially approved for heavily pretreated MM patients, followed by ciltacabtagene autoleucel (cilta-cel), both of which target BCMA on the surface of myeloma cells [202,203]. Ide-cel is composed of a mouse scFv (11D5-3) targeting domain, a 4-1BB (CD137) co-stimulatory domain, and a CD3ζ T-cell activation domain, and uses a lentivirus vector for CAR introduction into the genome of T cells [204,205]. On the other hand, cilta-cel is composed of two llama-derived variable heavy-chain-only (non-scFv) antigen recognition domains targeting two distinct regions of BCMA, a 4-1BB (CD137) co-stimulatory domain, and a CD3ζ T cell activation domain, and uses a lentivirus vector similar to ide-cel [206,207].

Despite the associated high responses, not all patients have durable responses after CAR T cell therapy [208], which is related to several tumor- and CAR-T-cell-construct-related factors [208]. Given that most CAR T cell products target BCMA, there is evidence suggesting that low baseline BCMA expression levels on tumor cells negatively impacts the efficacy of CAR T cells [209]. Additionally, myeloma cells can shed BCMA, leading to lower surface concentration and circulation of soluble BCMA (sBCMA). sBCMA binds to CAR T cells, blocking their interactions with BCMA on the surface of malignant cells, resulting in the decreased efficacy of CAR T cells, as shown in preclinical studies [210,211]. One mechanism that could explain antigenic loss is acquired biallelic BCMA deletion, resulting in decreased BCMA expression [212,213]. High tumor load also appears to negatively affect the efficacy of CAR T cell therapy, perhaps due to CAR T cell exhaustion [202,214]. High expression of immune checkpoints on the surface on myeloma and CAR T cells can also attenuate CAR T cell activity [193,215,216]. CAR T cells typically induce malignant cell death via the release of toxic granules containing perforin and serine proteases, and the induction of apoptosis via receptor cross-linking. It has been described that, in cases of treatment resistance, tumor cells were found to overexpress several antiapoptotic molecules including serine protease inhibitors or other proteins interfering with crosslinking [217,218].

The quality and composition of T cells in the leukapheresis product can also influence the outcomes of CAR T cell therapy. A high frequency of less-differentiated early memory T cells [219,220,221] and a high CD4/CD8 T cell ratio in the apheresis collection [201,221], which is typically seen in patients early in their disease course [222], leads to higher CAR T cell proliferation, expansion, and persistence and subsequently higher response rates [223,224]. On the contrary, multiple prior therapies in heavily pretreated patients are believed to negatively affect the fitness and constitution of the T-cell compartment.

There are several approaches to overcome these challenges, including dual-targeted CAR T cells harboring two different CAR or one CAR with two different antigen binding domains [225]. Several preclinical studies are currently investigating simultaneous targeting of BCMA and SLAMF7, GPRC5D, or CD38, molecules uniformly expressed on MM cells [217,225,226,227]. Another idea is to manufacture CAR T cells that can secrete checkpoint inhibitory antibodies such as anti-PD1 or anti-PD-L1 or CAR T cells in which genes that express immune checkpoints are knocked down [228,229,230]. Optimizing the structure of CAR by adding a costimulatory domains is also important as it can lead to improved persistence and activity with decreased exhaustion. [214,231,232]. In an effort to collect a more balanced T cell product, which would theoretically enhance CAR T cell function, allogeneic CAR T cells generated from the T cells of healthy donors have also been manufactured and assessed in the clinical context, with promising outcomes and an acceptable side effect profile [233].

### 5.7. Peptite Vaccines

An attractive approach for controlling tumors is developing synthetic peptide vaccines derived from widely expressed tumor-associated antigens (TAAs), which have the ability to bind multiple MHC class I and class II alleles, thus activating T-cell-mediated tumor destruction. This method is considered safe, and theoretically can be highly potent, specific, and long lasting. [234]. One approach is to target MUC1 (mucin 1, cell surface associated), a mucin-like glycoprotein highly expressed in a variety of epithelial and hematologic tumors including MM [235,236,237]. MUC1 is made of a large soluble extracellular alpha subunit containing the tandem repeats array (TRA) and a smaller beta subunit containing the transmembrane and cytoplasmic domains. The MUC1 signal peptide (SP) domain of the MUC1 binds multiple MHC class I and class II alleles, generating a robust T cell immunity; therefore, it was felt to serve as suitable vaccine candidate. Based on this rationale, a 21mer peptide vaccine encoding the complete signal domain of MUC1 was constructed and named as the ImMucin (VXL100) vaccine. [235,236,237]. In a phase 1/2 study, 15 MM patients were enrolled and vaccinated with ImMucin; however, only 9 patients completed the vaccination course (a total of six doses) [234]. ImMucin vaccination resulted in a significant increase in the percentage of both γ-interferon-producing CD4+ and CD8+ T cells in all patients. Additionally, a 9.4-fold increase in peripheral blood mononuclear cells and a 6.8-fold increase in anti-ImMucin antibodies was noted. Disease improvement or stability persisted for 17.5–41.3 months post-vaccination. These findings suggested a potential therapeutic benefit of ImMucin in MUC1-positive tumors in MM patients.

Similarly, PVX-410 is a human leukocyte antigen (HLA)-A2-restricted multipeptide vaccine for patients with SMM [238,239]. The vaccine is composed of a unique combination of four immunogenic peptides (XBP1_US184–192_, XBP1_SP367–375_, CD138_260–268_, and CS1_239–247_) derived from specific tumor target antigens (XBP1, CD138, and CS1, respectively) highly expressed on MM cells. These peptides were found to activate the immune system in an HLA-A2-specific manner, inducing antigen-specific CTLs against HLA-A2-positive MM cells. [239]. In a phase 1/2a study, 22 patients with SMM and the presence of HLA-A2 were divided into three groups, PVX-410 (low and target dose) or lenalidomide with PVX-410 [240]. In all cohorts, the PVX-410 vaccine induced a highly effective immune response against MM cells, with expansion of the CD3+ CD8+ CTL compartment against the XBP1, CD138, and CS1 antigenic epitopes. The response was further enhanced during treatment with lenalidomide. In the target-dose cohort, 1 out of 9 patients progressed (median TTP 36 weeks), as well as 1 out of 12 in the combination cohort (median TTP no reached).

A relatively recent phase 1 trial demonstrated the role of the PD-L1 peptide (IO103) vaccine in MM patients [241]. As previously mentioned, upregulation of PD-1/PD-L1 [121,242,243] is associated with poor prognosis in patients with MM [244]. Stimulation with the IO103 peptide stimulated PD-L1-specific T cytotoxic cells against PD-L1-expressing MM cells. In this study, 10 patients with MM who were 6 months post AHCT were enrolled [241]. Patients received vaccination with IO103 up to 15 times within one year. All patients showed a peptide-specific immune response in peripheral blood mononuclear cells and in skin-infiltrating lymphocytes. Three out of ten patients had improvement of response (over 100 days post-transplant) [241].

Quian and colleagues assessed the role of Dickkopf-1 (DKK1), a protein that is highly expressed in MM cells but not in normal tissues, as a potential vaccine candidate. Their in vitro experiments showed that cytotoxic T lymphocytes were able to recognize DKK1 peptides naturally presented by MM cells in the context of HLA-A*0201 molecules. This led to the immune-mediated destruction of MM cells, hence suggesting that DKK1 could be a potentially important antigen for immunotherapy in MM [245]. Further experiments from the same group in mouse murine myeloma models showed that vaccination with DKK1-DNA not only prevented mice from developing MM, but was also therapeutic against active MM. DKK1 vaccination elicited strong DKK1- and tumor-specific CD4+ and CD8+ immune responses, providing extra evidence for targeting DKK1 in MM patients [246]. Despite these encouraging outcomes, vaccination against MM has not been adopted in the clinical setting. There is currently an ongoing pilot phase 1 study exploring the application of the DKK1 vaccine in patients with MGUS and stable or smoldering myeloma (NCT03591614).

## 6. Targeted Therapies and Small Molecules

### 6.1. Exportin Inhibitors

The nuclear pore complexes (NPC) are large cylindrical channels, composed of several copies of >30 different proteins called nucleoporins [247]. The main function of the NPC is to fuse the inner and outer nuclear membranes, enabling traffic of vital macromolecules between the nucleus and the cytoplasm, a process which is mediated by specific protein carriers, importins and exportins. [248]. Exportin-1 (XPO1) is one of the most well-characterized nuclear exporters, involved in shuttling of multiple cargo proteins such as tumor suppressor proteins, cell cycle regulators, immune response regulators, and oncogenes, as well as mRNAs, out of the nucleus and into the cytoplasm, enhancing the synthesis of oncoproteins [249,250]. Overexpression of XPO1 leads to increased transfer of tumor suppressor and regulatory proteins into the cytoplasm, which further promotes cell proliferation and halts apoptosis, overall favoring carcinogenesis. Increased XPO1 levels have been observed in a variety of malignancies including CD138+ PCs from patients with active MM and are associated with poor survival outcomes, making XPO1 an attractive molecular target for novel therapies [251]. Selective inhibitors of nuclear export (SINE) are orally bioavailable small-molecule drugs that inhibit XPO1 by attaching to the binding site of the cargo, thus disrupting the nuclear–cytoplasmic trafficking. As a result, tumor suppressor proteins and regulators eventually accumulate in the nucleus, activating the apoptotic process and subsequently causing cell death [252,253].

Selinexor is an XPO1 inhibitor that has reduced the viability of MM cells in preclinical experiments, alone or in synergism with other anti-myeloma agents. [238] In detail, selinexor causes retention of tumor suppressor proteins in the nucleus such as p53, p27, and FOXO and decreases the levels of cell cycle promoters and antiapoptotic proteins, leading to cell cycle arrest, with subsequent caspase activation and cell death [254,255,256] (Figure 4). It has also been shown to block NF-kB, which regulates osteoclast differentiation [257]. It is currently approved in combination with bortezomib/dexamethasone or dexamethasone alone in the relapsed/refractory setting, with ongoing trials investigating different combinations with other novel agents [258].

### 6.2. Histone Deacetylase Inhibitors

Histone deacetylase (HDAC) inhibitors act at the epigenetic level, by removing acetyl groups from mainly the histones, proteins forming the nucleosome, which play a critical role in chromatin organization [259] (Figure 4). In MM, overexpression of HDAC, especially HDAC-1, has been associated with a poor prognosis and with resistance to PI [260,261]. Panobinostat and vorinostat are pan-HDAC inhibitors leading to blockade of disposal of several proapoptotic proteins through the unfolded protein response, disrupting protein homeostasis and resulting in cell death via apoptosis [262].

Panobinostat was initially approved by the FDA; however, due to lack of confirmatory post-approval clinical studies, required as part of the accelerated approval process, it was withdrawn from the market. Other experimental HDAC inhibitors, such as quisinostat, CUDC-907, and AR-42, have also been studied in the pre- and clinical settings.

### 6.3. BCL2 Inhibitors

The B cell lymphoma-2 (Bcl-2) protein family consists of pro- and anti-apoptotic proteins which regulate the intrinsic pathway of apoptosis. Bcl-2 is an anti-apoptotic protein of the Bcl-2 family containing four homogeneous domains called BH1, BH2, BH3, and BH4, whereas pro-apoptotic proteins in the same family only contain the BH3 domain and are called BH3-only proteins [263]. The latter subcategory primarily works by binding to anti-apoptotic proteins, activating the BAX/BAK proteins, directly or indirectly, and inducing apoptosis [264,265]. Overexpression of the anti-apoptotic Bcl-2 has primarily been observed in the subgroup of MM patients harboring the translocation of the chromosomes 14 and 17 [266]. High levels of Bcl-2 promote cell survival and tumorigenesis and have been associated with poor outcomes and resistance to traditional anti-myeloma agents; therefore, Bcl-2 represents an attractive target for novel therapies.

Venetoclax is an orally bioavailable BH-3 mimetic that selectively inhibits Bcl-2, disrupting the anti-apoptotic pathway, thus favoring cell death in a TP-53-independent manner. (Figure 4) Venetoclax is particularly efficacious in the subset of MM patients with the translocation (11;14). These patients express high levels of Bcl-2, possibly due to increased tumoral dependence upon Bcl-2 [267]. As a result, translocation (11;14) has emerged as the first predictor of susceptibility to Bcl-2 inhibition in MM patients [268]. Venetoclax is not FDA-approved yet; however, the NCCN guidelines recommend its use in RRMM with the translocation (11;14).

### 6.4. Hypomethylating Agents

While hypomethylating agents have been effective for the treatment of myeloid leukemia, it seems they had limited efficacy in a phase 1b trial of 42 heavily pretreated patients with RRMM. This trial assessed the addition of Azacytidine, a DNA methylation inhibitor, to lenalidomide and dexamethasone with the purpose of overcoming refractoriness to IMiD via interfering with pathways associated with PC differentiation, apoptosis, and immune recognition. The overall response rate was 32%, with 10% achieving very good partial response, and the median PFS was 3.1 months. The levels of the azacytidine-inactivating enzyme cytidine deaminase (CDA) were measured to assess any potential correlation with treatment response, and it was found that low plasma CDA levels were associated with greater clinical benefit [269]. Currently, there is an ongoing phase II trial evaluating azacitidine in combination with daratumumab and dexamethasone in patients with RRMM who have already received daratumumab (NCT04407442).

### 6.5. Proteolysis-Targeting Chimera

Proteolysis-targeting chimera (PROTAC) is a class of bi-functional degrader molecules that have designed to selectively target and then degrade intractable cellular proteins via activation of the ubiquitin–proteasome system. These molecules typically consist of two ligand-binding domains, one that binds to a E3 ubiquitin ligase and another that binds a protein of interest (POI) [270]. The two domains are connected through a linker. PROTAC ultimately forms a complex between an E3 ligase and POI, which results in ubiquitination and subsequent degradation by the proteasome. The domain of PROTAC binding to an E3 ligase can be either a phthalimide derivative binding to a cereblon (CRL4 CRBN) E3 ligase (Cereblon PROTAC), or a von Hippel–Lindau (VHL) binding to VHL E3 ligase (VHL PROTAC) [270,271].

An initial PROTAC called dBET1 was constructed using thalidomide (as an E3 ligase binding domain) and JQ1 (as a POI binding domain) [272]. JQ1 is a small molecule binding to Bromodomain-Containing Protein 4 (BRD4). This PROTAC induced cereblon-dependent degradation of BRD4 and subsequent down-regulation of MYC, leading to the cytotoxicity of AML cells [273]. In vitro and in vivo pre-clinical studies in MM models using PROTAC targeting BRD4 and other BET proteins reduced the viability of MM cell lines in a time- and concentration-dependent manner and demonstrated suppressed MYC and Akt/mTOR signaling [274]. PROTAC was able to overcome resistance to PI and IMiD, and their activity was maintained in MM cells with wild-type or deleted TP53. Further studies demonstrated that BET-targeted PRTOAC was able to inhibit cell proliferation of multiple human-derived MM cell lines and fresh myeloma samples and suggested potential synergy with systemic agents including selinexor [275]. Another, newer experimental PROTAC targeting the proteasome substrate receptor hRpn13, which was found to be upregulated in MM, was tested in in vitro studies [276]. Optimization of PROTAC design for potential clinical development is eagerly awaited.

## 7. Emerging Approaches and Future Directions

### 7.1. Protein Disulfide Isomerase 1 Inhibitors

In MM, the malignant PCs exhibit very high protein synthesis burden, as evident by the abundant secretion of non-functional monoclonal antibodies and cytokines [277]. This requires re-arrangement of intramolecular disulfide bonds mediated by the protein disulfide isomerase (PDI) family in the endoplasmic reticulum (ER), an organelle that is responsible for the biosynthesis, folding, maturation, stabilization, and trafficking of transmembrane and secretory proteins [278,279]. PDIA1 is the main endoplasmic reticulum resident isoform of this multifunctional protein family, overexpression of which has been observed in a variety of cancers [280,281,282]. As previously mentioned, malignant PCs are highly dependent on proteostasis, largely taking place in the ER [283]. Disruption of the ER, which constantly functions at maximum capacity in MM cells, can cause dysregulation of the multi-level protein folding process, known as ER stress, secondary to the accumulation of unfolded or misfolded proteins, leading to activation of the unfolded protein response (UPR) and re-establishment of proteostasis and ER recovery [284]. However, in cases of severe ER stress, damage is irreversible and cell apoptosis is inevitable (Figure 5) [285].

Disruption of proteostasis has proven to be overwhelmingly efficacious in MM with the use of PI, suggesting that inhibition of proteostasis at earlier levels, alone or in combination with PI, may result in higher efficacy, especially in patients with PI-refractory disease [286]. Recently, analysis of bone marrow specimens from 690 individuals showed high expression of PDIA1 in RRMM patients that was inversely related with overall survival [287]. This observation suggested that upregulation of PDIA1 expression in patients with RRMM may confer an adaptive mechanism of the relapse/refractory state and, in part, be responsible for the disease resistance to known therapies, hence indicating that PDIA1 is a good option for targeted therapy development.

CCF642 was the first PDIA1 inhibitor constructed in the lab, which leads to irreversible lethal ER stress with subsequent MM cell apoptosis in in vitro and in vivo preclinical studies of bortezomib-resistant MM cells, without toxic effects on the healthy bone marrow hematopoietic cells [287]. However, due to poor solubility and lack of availability, it was not advanced to clinical studies. Later, a pan-PDI inhibitor, E64FC26, was tested in mice and was found to improve survival and enhance the activity of bortezomib without any adverse effects [288]. Most recently, a new, more selective PDIA1 inhibitor, CCF642-34, was developed, with improved solubility, selectivity, and potency [286].

At a molecular level, inhibition of PDIA1 with CCF642-34 inducted irreversible ER stress and cell death via PARP activation and caspase 3 cleavage, as well as increased reactive oxygen species (ROS) production, leading to upregulation of the NRF2 pathway in MM cells. Analysis of RNA NGS from MM cells demonstrated selectivity in ER stress induction, particularly via activation of UPR, ER-associated protein degradation (ERAD), and the ATF6/PERK pathway. Interestingly, bortezomib-resistant cell lines were noted to be more sensitive to the inhibition with CCF642-34 compared to bortezomib-naïve cells, which further supported the hypothesis of increased dependence of bortezomib-resistant MM cells on PDIA1. Further experiments showed synergism of CCF642-34 with bortezomib [286]. The CCF642-34 was then evaluated on a mouse xenograft with 50% of mice surviving beyond the end of the experiment (180 days). These extensive preclinical experiments highlighted the potential of PDIA1 inhibitors and further supported the strategy of targeting the PDIA1 isoform for the treatment of RRMM.

### 7.2. Peptidylprolyl Isomerase A

A recent single-arm phase 2 study (NCT04065789), assessed the combination of carfilzomib, daratumumab, lenalidomide, and dexamethasone in MM patients who failed to respond or experienced early relapse after a bortezomib-based induction regimen [289]. Single-cell RNA sequencing was applied to the participants’ samples, in order to explore the molecular pathways of MM resistance. The 41 MM patients of the trial were compared to 11 healthy individuals and 15 NDMM patients. Results showed upregulation of genes associated with endoplasmic reticulum stress pathways, such as peptidylprolyl isomerase A (PPIA), in patients with relapsed/refractory disease [289]. Cyclophilin A or PPIA is a critical enzyme in the protein folding response pathway, upregulation of which may represent an escape mechanism, and, therefore, the authors concluded that it can be a potential new therapeutic target for future drug development. Deletion of the PPIA gene or inhibition of PPIA with the use of a small molecule inhibitor, called ciclosporin, significantly sensitized the MM cells to PI.

### 7.3. Sec61 Translocon

Translocon is a protein of the membrane of the ER which mediates membrane insertion of most membrane proteins of organelles and facilitates translocation of almost all newly synthesized polypeptides destined for organelles, as well as most precursors of secretory proteins [290]. Sec61 is polypeptide-conducting channel, which is part of the translocon along with other proteins. Targeting the Sec61 of the translocon, thus disrupting transport of newly secreted or transmembrane proteins into the ER, leads to their degradation by the proteasome, which can eventually lead to cell apoptosis due to continuous proteotoxic stress response. The use of mycolactone, a small molecule that inhibits Sec61, along with PI was found to be effective in inducing apoptosis and synergistic in pre-clinical models of laboratory-derived MM cell lines [291]. This combination was found to cause enhanced ER stress with activation of pro-apoptotic UPR in MM cells, as reflected on transcript as well as protein levels. The synergy was maintained in bortezomib-resistant MM.1S cell lines, as well as in patient-derived MM cells. This target should be further explored as a potential novel therapeutic approach and drug development [291].

### 7.4. Cyclin-Dependent Kinase 6 (CKD6)

A recently published study performed integrated global quantitative tandem-mass-tag (TMT)-based proteomic and phosphoproteomic analyses in MM samples to identify a non-genetic resistance mechanism for IMiD that can be targeted by pharmacologic intervention. The results showed upregulation of CDK6, which appeared to decrease sensitivity to IMID [292]. Inhibition of CDK6 with palbociclib or degradation of CDK6 by PROTAC was found to be highly effective and synergistic with IMiD in pre-clinical in vivo and in vitro models [292]. These findings suggested further investigation of CDK6 as a potential drug target.

## 8. Conclusions

Despite the progress made to advance the therapeutic modalities in MM, treatment resistance and relapse continue to be a major issue. Hence, novel targeted and safer treatments are needed to combat this incurable disease. Understanding the genetic and epigenetic basis of the molecular evolution of MGUS to MM is key. For this purpose, using the power of next-generation sequencing coupled with genome editing will not only enable better prognostication, risk stratification, and prediction of therapy response, but, most importantly, will lead to the discovery of new strategies for prevention as well as curing of MM.

## Figures and Tables

**Figure 1 cancers-14-04082-f001:**
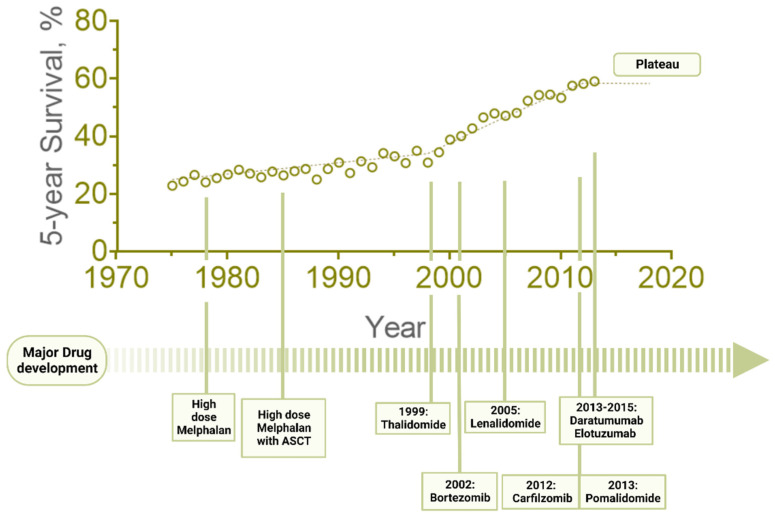
**Five-Year Relative Survival Percent, as reported at the Surveillance, Epidemiology, and End Results (SEER) Program***(SEER Cancer Statistics Review, 1975–2018, National Cancer Institute. Bethesda, MD, USA*, https://seer.cancer.gov/csr/1975_2018/ (accessed on 22 July 2022), *based on November 2020 SEER data submission, posted to the SEER web site)*. The overall survival of MM dramatically improved after the development of novel proteasome inhibitors, immunomodulators and monoclonal antibodies. However, the survival advantage has plateaued over the past several years, emphasizing the need for discovery of new therapeutic modalities.

**Figure 2 cancers-14-04082-f002:**
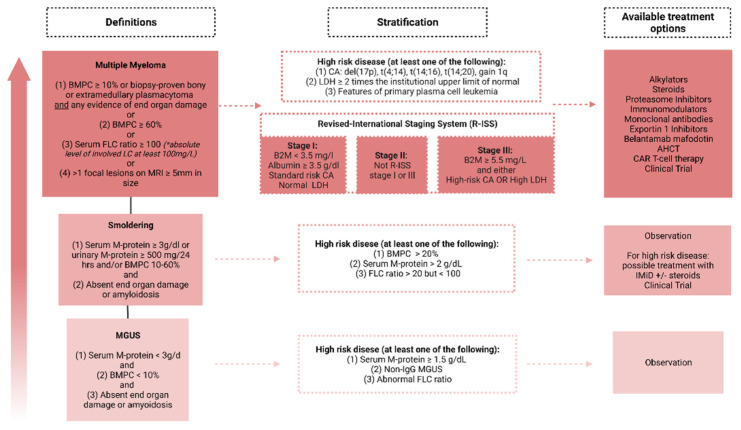
**Clinical definitions, risk stratification, and available therapies for multiple myeloma and preexisting plasma cell disorders.** End organ damage is defined as: Hypercalcemia (serum calcium > 1 mg/dL higher than the upper limit of normal or >11 mg/dL); renal insufficiency, creatinine clearance < 40 mL/minute or serum creatinine > 2 mg/dL; anemia with a hemoglobin value > 2 g/dL below the lowest limit of normal or a hemoglobin value < 10.0 g/dL; and Bone lesions, ≥1 osteolytic lesion on imaging. Abbreviations: M, monoclonal; BMPC, Bone Marrow Plasma Cells; FLC, Free Light Chain; B2M, Beta-2 Microglobulin; CA, Cytogenetics, IMiD, Immunomodulators; AHCT, Autologous Hematopoietic Cell Transplantation; CAR, Chimeric Antigen Receptor.

**Figure 3 cancers-14-04082-f003:**
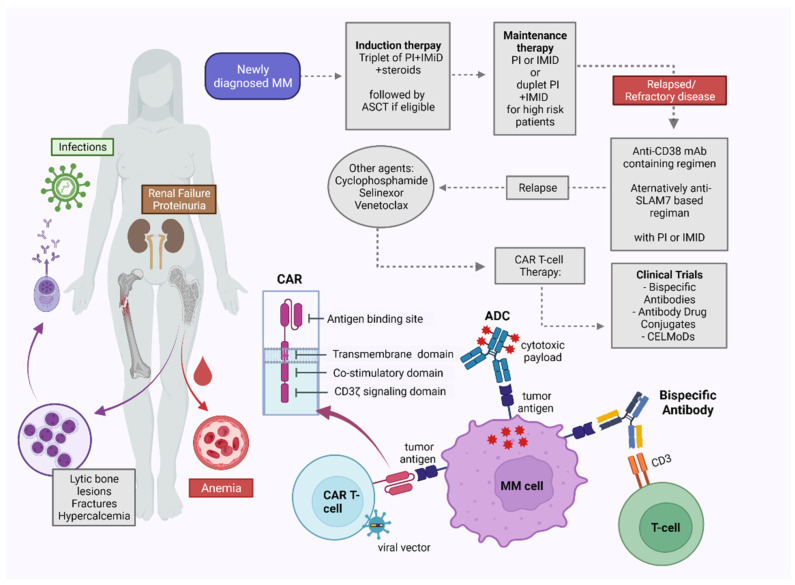
**The basic principles of the immunotherapies in MM and their place in the current treatment landscape.** CAR T cells are T cells genetically modified with the use of a viral vector to express a chimeric antigen receptor on their surface, which targets specific tumor antigens of malignant plasma cells. Similarly, bispecific antibodies are monoclonal antibodies targeting both an antigen on the malignant MM cells and simultaneously an antigen on the surface of physiologic T cells, creating an immunologic bridge. ADC are monoclonal antibodies against antigenic epitopes on the surface on MM cells, carrying a cytotoxic payload. The binding of the above agents to their antigenic targets on malignant MM cells leads to activation of the immune system, with subsequent destruction of the MM cells.

**Figure 4 cancers-14-04082-f004:**
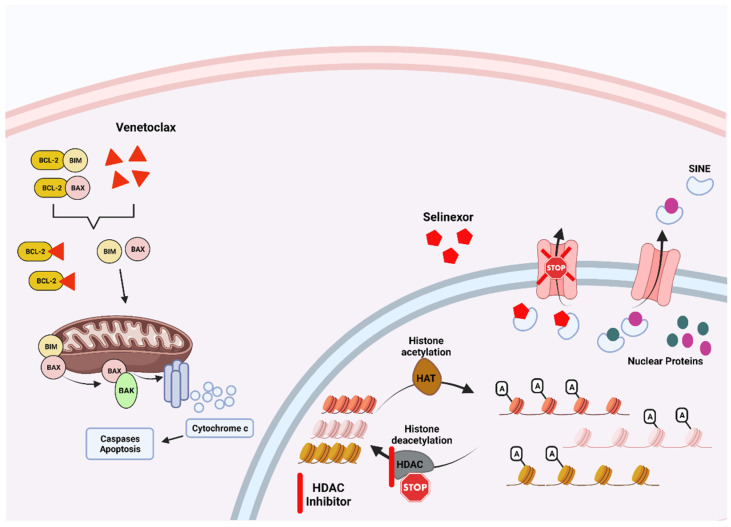
**Underlying mechanism of action of selected targeted therapies used for MM in the clinical setting.** Venetoclax works by primarily binding to the BCL-2 anti-apoptotic protein, allowing the activation of BAK and subsequently caspases leading to MM apoptosis. Selinexor blocks the transport of vital proteins and other molecules from the nucleus to the cytoplasm of the MM cells, leading to cell death. Histone deacetylation inhibitors act at an epigenetic level, blocking the deacetylation of the DNA in the nucleus of the malignant cell.

**Figure 5 cancers-14-04082-f005:**
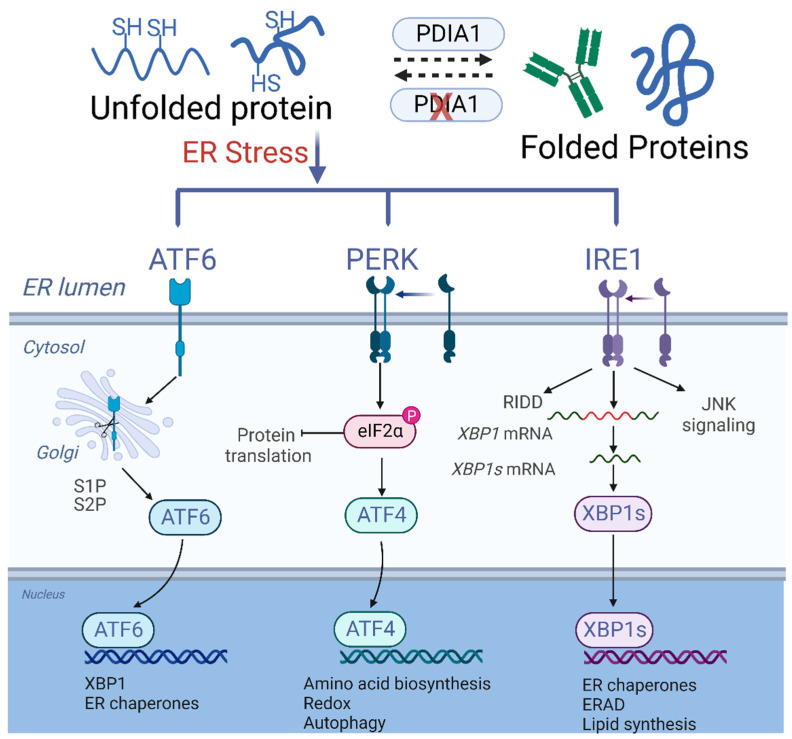
**Heightened PDIA1 is critical to maintain ER homeostasis in MM Cells.** Disruption of the ER causes dysregulation of the protein folding process, known as ER stress, secondary to the accumulation of unfolded or misfolded proteins. This leads to activation of the unfolded protein response (UPR), with subsequent activation of downstream pathways inducing apoptosis. Abbreviations: PDIA1, protein disulfide isomerase inhibitor A1; ER, endoplasmic reticulum; UPR, unfolded protein response.

**Table 1 cancers-14-04082-t001:** Major clinical trials of the standard-of-care regimens for MM currently used in routine clinical practice.

Regimen	Trial Name NCT Number	Phase	N	Disease Status	Outcomes
**Dara-Vd vs. Vd**	CASTOR NCT02136134	3	498	RRMM	**ORR:** 83.8% vs. 63.2% (*p* < 0.0001) **≥CR:** 28.8% vs. 9.8% (*p* < 0.0001) **m-PFS:** 16.7 vs. 7.1 months (*p* < 0.0001)
**Dara-Rd** **vs. Rd**	POLLUX NCT02076009	3	569	RRMM	**ORR:** 92.9 vs. 76.4% (*p* < 0.0001) **≥CR:** 56.6 vs. 23.2% (*p* < 0.0001 **m-PFS:** 44.5 vs. 17.5 months (*p* < 0.0001)
**Dara-Kd** **vs. Kd**	CANDOR NCT03158688	3	446	RRMM	**ORR:** 84% vs. 75% (*p* = 0.0080) **≥CR:** 33% vs. 13% **m-PFS:** 28.6 vs. 15.2 months (*p* < 0.0001)
**Dara-Pd** **vs. Pd**	APOLLO NCT03180736	3	304	RRMM	**ORR:** 69% vs. 46% (*p* < 0.0001) **≥CR:** 25 vs. 4%, (*p* < 0.0001) **m-PFS:** 12.4 vs. 6.9 months (*p* = 0.0018)
**Dara-Rd** **vs. RD**	MAIA NCT02252172	3	737	NDMM	**ORR:** 92.9% vs. 81.6 (*p* < 0.0001) **≥CR:** 51% vs. 30% (*p* < 0.0001) **m-PFS:** NR vs. 34.4 months (*p* < 0.0001)
**Dara-VMP vs. VMP**	ALCYONE (NCT02195479)	3	706	NDMM	**ORR:** 90.9% vs. 73.9% (*p* < 0.0001) **≥CR:** 46% vs. 25% (*p* < 0.0001) **m-PFS:** 36.4 vs. 19.3 months (*p* < 0.0001)
**Dara-VTd** **vs. VTd**	CASSIOPEIA NCT02541383	3	1085	NDMM	**At day 100 post AHCT:** **CR:** 29% vs. 20% (*p* = 0.001) **≥CR:** 39% vs. 26% (*p* < 0.0001) **m-PFS:** NR for both groups (*p* < 0.0001)
**Dara-VRd** **vs. VRd**	GRIFFIN NCT03710603	2	207	NDMM	**At the end of post-AHCT consolidation:** **ORR:** 99% vs. 91.8 (*p* = 0.016) **sCR:** 42.4% vs. 32% (*p* = 0.068) **m-PFS:** NR for both groups
**Isa-Pd vs. Pd**	ICARIA NCT02990338	3	307	RRMM	**ORR:** 60% vs. 35% (*p* < 0.0001) **m-PFS:** 11.5 vs. 6.5 months (*p* = 0.001) **m-OS:** 24.6 vs. 17.7 months (*p* = 0.028)
**Isa-Kd vs. Kd**	IKEMA NCT03275285	3	302	RRMM	**CR:** 40% vs. 28% **≥VGPR:** 73% vs. 56% (*p* = 0.0011) **m-PFS:** 19.15 months vs. NR (*p* = 0.0007)
**Elo-Rd** **vs. Rd**	ELOQUENT-2 NCT01239797	3	321	RRMM	**ORR:** 79%, vs. 66% (*p* < 0.001) **m-PFS:** 19.4 vs. 14.9 months (*p* < 0.001) **m-OS:** 48.3 vs. 39.6 months (*p* = 0.0408)
**Elo-Pd** **vs. Pd**	ELOQUENT-3 NCT02654132	2	117	RRMM	**ORR:** 53% vs. 26% **m-PFS:** 10.3 vs. 4.7 months (*p* = 0.008) **m-OS:** 29.8 vs. 17.4 months (*p* = 0.0217)

Abbreviations: dara, daratumumab; Vd, bortezomib–dexamethasone; Rd, lenalidomide–dexamethasone; Kd, carfilzomib–dexamethasone; Pd, pomalidomide–dexamethasone; VMP, bortezomib–melphalan–dexamethasone; VTd, bortezomib–thalidomide–dexamethasone; VRd, bortezomib–lenalidomide–dexamethasone; Isa, isatuximab; m-PFS, median progression-free survival; m-OS, median overall survival.

**Table 2 cancers-14-04082-t002:** Major CAR T cell and Bispecific Antibody Clinical Trials with reported outcomes in MM.

Agent Name	Target Antigen	NCT Number (Trial Name)	Phase	N	Disease Status	Outcomes	CRS ICANS
**CAR -T Cell Therapy Products**							
**bb21217**	BCMA	NCT03274219 (CRB-402)	1	46	RRMM	▪ ORR: 55%, ≥CR: 18%, VGPR: 30% ▪ mPFS: 7.2 months	▪ CRS: 67% ▪ ICANS: 22%
**Idecabtagene vicleucel**	BCMA	NCT02658929 (CRB-401)	1	62	RRMM	▪ ORR: 76%, ≥CR: 39%, ≥VGPR: 65% ▪ mPFS/mOS: 8.8/34.2 months	▪ CRS: 76% ▪ ICANS: 44%
		NCT03361748 (KarMMa)	2	128	RRMM	▪ ORR: 73%, ≥CR: 33%, ≥VGPR: 52% ▪ mPFS/mOS: 8.8/19.4 months	▪ CRS: 84% ▪ ICANS: 18%
**Ciltacabtagene autoleucel**	BCMA	NCT03548207 (CARTITUDE-1)	1/2	113	RRMM	▪ ORR: 98%, CR: 82.5%, VGPR: 95% ▪ mPFS and mOS: NR	▪ CRS: 95% ▪ ICANS: 21%
		NCT04133636 (CARTITUDE-2)	2	20	RRMM	▪ ORR: 95%, CR: 75%, ≥VGPR: 85% ▪ mPFS and mOS: NR	▪ CRS: 85% ▪ ICANS: 20%
**Zevorcabtagene autoleucel**	BCMA	NCT03716856 NCT03302403 NCT03380039	1	24	RRMM	▪ ORR: 87.5%, ≥CR: 80% ▪ mPFS 9.2 months	▪ CRS: 62.5% ▪ ICANS: 12.5%
		NCT03975907 (LUMMICAR-1)	1/2	38	RRMM	▪ ORR: 92%, CR: 79% ▪ mPFS: 22.7 months	▪ CRS: 73.7%
		NCT03915184 (LUMMICAR-2)	1/2	34	RRMM	▪ ORR: 100% ▪ mPFS and mOS: NR	▪ CRS: 86%
**Orvacabtagene autoleucel (JCARH125)**	BCMA	NCT03090659 (LEGEND-2)	1	74	RRMM	▪ ORR 88%, CR 73% ▪ mPFS: 18 months, mOS: NR	▪ CRS: 92% ▪ ICANS: 21%
		NCT03430011 (EVOLVE)	1/2	115	RRMM	▪ ORR: 82% ▪ mPFS: NR	▪ CRS: 75%
**CT103A**	BCMA	NCT05066646 (FUMANBA-1)	1/2	79	RRMM	▪ ORR: 94.9% ▪ CR/sCR: 69.6%	▪ CRS 94.9% ▪ ICANS: 2.5%
**CART-ddBCMA**	BCMA	NCT04155749	1	25	RRMM	▪ ORR: 100%, ≥CR: 75% ▪ mPFS and mOS: NR	▪ CRS: 100% ▪ ICANS: 16%
**GC012F**	BCMA/CD19	NCT04236011 NCT04182581	1	28	RRMM	▪ ORR: 80–100%	▪ CRS: 100%
**OriCAR-017**	GPRC5D	NCT05016778	1	11	RRMM	▪ ORR: 100% ▪ MRD (10^−5^) negativity: 100%	▪ CRS: 100%
**Bispecific Antibodies**							
**Teclistamab**	BCMAxCD3	NCT03145181 NCT04557098 (MajesTEC-1)	1/2	165	RRMM	▪ ORR: 63%, ≥CR 39.4% ▪ MRD (10^−5^) negativity: 26.7% ▪ mPFS: 11.3 months	▪ CRS: 72% ▪ ICANS: 3%
**Teclistamab + daratumumab**		NCT04108195 (TRIMM-2)	1b	46	RRMM	▪ ORR: 78% ▪ ≥VGPR: 73%	▪ CRS: 61% ▪ ICANS: 2.1%
**Elranatamab**	BCMAxCD3	NCT03269136 (MagnetisMM-1)	1	55	RRMM	▪ ORR: 64%	▪ CRS: 67%
		NCT04649359 (MagnetisMM-3)	2	60	RRMM	▪ ORR: NR	▪ CRS 58.9% ▪ ICANS: 3.6%
**Talquetamab**	GPRC5DxCD3	NCT03399799 (MonumenTAL-1)	1	174	RRMM	▪ ORR 63% ▪ ≥VGPR 50%	▪ CRS 79% ▪ ICANS: 7%

Abbreviations: ORR, overall response rate; CR, complete response; VGPR, very good partial response rate; mPFS, median progression-free survival; mOS, median overall survival; RRMM, relapsed/refractory multiple myeloma; CRS, cytokine release syndrome; NR, not reached.

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
