# Peer review of "Multiple Myeloma Therapy: Emerging Trends and Challenges"

_cancers, 2022, doi:10.3390/cancers14174082_

Round 1
Reviewer 1 Report
I read with interest the manuscript entitled "Multiple Myeloma Therapy: Emerging Trends and Challenges" by Dima et al.
This is a comprehensive review of the current knowledge on myeloma and its treatment and the potential future opportunities. The manuscript is well written and includes all the updated vital information for a scientist in the field. There are only minor issues:
1- Page 6, lines 178-179: this statement should be reconsidered carefully since MICA is an activating ligand for NKG2D and usually its downregulation (proteolytic, exosome-mediated, etc,) is associated with immune escape and suppression; however the authors are reporting the opposite.
2- Page 6, line 213: The statement is ambiguous and may be misleading to the author. Needs to be revised.
3- There are frequent punctuation errors throughout the manuscript, too frequent to mention.
4- There are some language errors that need to be addressed.
Author Response
There are only minor issues:
1- Page 6, lines 178-179: this statement should be reconsidered carefully since MICA is an activating ligand for NKG2D and usually its downregulation (proteolytic, exosome-mediated, etc,) is associated with immune escape and suppression; however the authors are reporting the opposite.
Answer: Thank you for your suggestions. We have modified the sentence, page 7, line 302.
2- Page 6, line 213: The statement is ambiguous and may be misleading to the author. Needs to be revised.
Answer: modified/removed per reviewer’s suggestion.
3- There are frequent punctuation errors throughout the manuscript, too frequent to mention.
Answer: We have corrected all punctuation errors. Thank you for careful reading.
4- There are some language errors that need to be addressed.
Answer: Addressed
Reviewer 2 Report
This review paper summarized the current therapeutic strategies for multiple myeloma and discussed the future promising approaches. The topic is important to the basic research in multiple myeloma as well as to the clinical studies for the treatment of multiple myeloma. This paper is well-organized, while the manuscript is very similar (coverage, even several head/subhead titles) with one recently published, and the references for the background are not enough, especially about the current review papers with the same/similar topic.
The authors omitted the very similar review paper, which has almost all the same/similar subtitles. (Yang, et al. Emerging agents and regimens for multiple myeloma. J Hematol Oncol, 2020, 13, 150.). The current manuscript has many sections (e.g., Immunomodulatory Drugs, Proteasome Inhibitors, Monocloal Antibodies, Immune Checkpoint Inhibitors, Antibody Drug Conjugates, Bispecific Antibodies, Bispecific T-cell Engagers, Chimeric Antigen Receptor (CAR) T-cell Therpay, Exportin Inhibitors, BCL2 Inhibitors, and Hypomethylating Agents), which are covered by Yang, et al paper with exactly same or same-meaning subtitles.
Major points:
This review paper is very similar with the published paper in 2020 as mentioned in the general comments. The authors are suggested to emphasize more on the mechanism studies to make some differences for this manuscript. Though the figures in the manuscript show mechanisms, the discussion is very less about the mechanisms. Most of words are about the therapies and effects, similar with those in the above-mentioned paper.
Minor points:
1. The tables for sections 3.1 Immunomodulatory Drugs, 3.2 Proteasome Inhibitors, 4.2 Naked Monoclonal Antibodies, 4.4 Antibody Drug Conjugates, and 4.6 Chimeric Antigen Receptor (CAR) T-cell Therapy, are required for the summary (including the clinical trial numbers) to improve the readability of this manuscript.
2. The vaccines for multiple myeloma cells should be discussed in the section of immunotherapy.
Author Response
Reviewer 2:
This review paper summarized the current therapeutic strategies for multiple myeloma and discussed the future promising approaches. The topic is important to the basic research in multiple myeloma as well as to the clinical studies for the treatment of multiple myeloma. This paper is well-organized, while the manuscript is very similar (coverage, even several head/subhead titles) with one recently published, and the references for the background are not enough, especially about the current review papers with the same/similar topic.
The authors omitted the very similar review paper, which has almost all the same/similar subtitles. (Yang, et al. Emerging agents and regimens for multiple myeloma. J Hematol Oncol, 2020, 13, 150.). The current manuscript has many sections (e.g., Immunomodulatory Drugs, Proteasome Inhibitors, Monoclonal Antibodies, Immune Checkpoint Inhibitors, Antibody Drug Conjugates, Bispecific Antibodies, Bispecific T-cell Engagers, Chimeric Antigen Receptor (CAR) T-cell Therapy, Exportin Inhibitors, BCL2 Inhibitors, and Hypomethylating Agents), which are covered by Yang, et al paper with exactly same or same-meaning subtitles.
Major points:
This review paper is very similar with the published paper in 2020 as mentioned in the general comments. The authors are suggested to emphasize more on the mechanism studies to make some differences for this manuscript. Though the figures in the manuscript show mechanisms, the discussion is very less about the mechanisms. Most of words are about the therapies and effects, similar with those in the above-mentioned paper.
Answer: Thank you for this feedback. We looked into the review article suggested by the reviewer. We believe that the purpose of the two reviews is very different. We have attempted a comprehensive approach to understand the existing therapy and their limitation along with emerging therapies. As the reviewer noted, we have put forward the comprehensive list of existing therapies and their reported mechanism of action which are not the part of other review. The content of the two articles is very different. The above-mentioned paper is clinically heavy, whereas our paper focuses on pathophysiologic and drug mechanism. Some section titles such as “monoclonal antibodies”, or “immunotherapy” are logically similar as these are uniformly used terms to describe specific drug classes and subclasses – no alternative terms can be used in some instances.
Minor points:
- The tables for sections 3.1 Immunomodulatory Drugs, 3.2 Proteasome Inhibitors, 4.2 Naked Monoclonal Antibodies, 4.4 Antibody Drug Conjugates, and 4.6 Chimeric Antigen Receptor (CAR) T-cell Therapy, are required for the summary (including the clinical trial numbers) to improve the readability of this manuscript.
Answer: We have added the tables with relevant link to clinical trials in the revised manuscript. Table 1: page 9 and Table 2: page 18 containing major trials were added.
- The vaccines for multiple myeloma cells should be discussed in the section of immunotherapy.
Answer: Please see “peptide vaccines” section added in the revised manuscript
Reviewer 3 Report
In the present review article, the authors have highlighted a comprehensive synopsis of the current and future of the multiple myeloma therapy. They have discussed on the therapeutic potential of emerging preclinical research in multiple myeloma.
The article is very well written, descriptive and comprehensive with the current knowledge on the therapeutic challenges and drug applications in multiple myeloma.
There are few points that can be addressed to strengthen the present form.
1. Discuss on whether the treatment regimens depend on the patient age and vulnerability.
2. Mention the challenges and failures associated with the current therapy in multiple myeloma.
3. It will be suggested to provide a table for the drugs already in the phase1/2/3 clinical trials in multiple myeloma and their outcomes.
4. The authors should acknowledge the site used for preparing the figures.
5. Discuss on whether the therapies discussed can show efficacy in patients with relapsed or refractory multiple myeloma.
Author Response
Reviewer 3:
- Discuss on whether the treatment regimens depend on the patient age and vulnerability.
Answer: Please see new section added “Factors influencing Treatment Strategy and Current Challenges” Page 4, line 138-177.
- Mention the challenges and failures associated with the current therapy in multiple myeloma.
Answer: Please see new section added “Factors influencing Treatment Strategy and Current Challenges” Page 4, line 138-177.
- It will be suggested to provide a table for the drugs already in the phase1/2/3 clinical trials in multiple myeloma and their outcomes.
Answer: We have added the tables with relevant link to clinical trials in the revised manuscript. Table 1: page 9 and Table 2: page 18 containing major trials were added.
- The authors should acknowledge the site used for preparing the figures.
Answer: Acknowledgement of the website bio-render was added at the end of the manuscript at the “Acknowledgement” section.
- 5. Discuss on whether the therapies discussed can show efficacy in patients with relapsed or refractory multiple myeloma.
Answer: Efficacy and response rates of available/experimental therapies were added to pertinent tables.
Reviewer 4 Report
This review briefly introduces the pathological progression of multiple myeloma (MM) and summarizes the current therapeutic advances in MM. The paper is well organized, neatly phrased, and supported with beautiful and informative figures. It offers an updated picture of emerging and promising therapeutic options in treating incurable MM.
Some concerns or comments for the current version:
1. The authors did not mention the traditional chemotherapies approved for MM, although this is not the focus of the review according to the title. However, as a comprehensive review, the manuscript should at least contain a short paragraph or a table illustrating what chemotherapies have been used for MM patients, such as alkylating agent.
2. In the section of immunotherapy, it is worthy to introduce the peptide vaccines (including but not limited to, ImMucin, PVX-410, and PD-L1 peptide IO103) that had been tested in clinical trials of MM patients.
3. In the section of “5. Targeted Therapies and Small Molecules”, the authors elaborated different categories of targeted inhibitors by introducing the mechanisms and clinical significance. One thing to add is the emerging technique of PROTAC (proteolysis-targeting chimera), which represents a novel category of small molecule to degrade the targets, instead of merely inhibiting the targets. There are several review or research articles discussing the applications of PROTAC molecules in MM treatment. This is a non-neglectable trend in MM therapy.
4. Regarding the heterogeneity of MM cells, scientists have already applied single cell sequencing to identify resistance pathways and therapeutic targets (e.g. Nat Med 2021, 27: 491-503). The authors mentioned “using the power of next generation sequencing coupled with genome editing” in conclusion remarks, but it could be better if they add one section before the conclusion to elaborate one or two studies.
5. The MM research community are keeping proposing potential targets, so it is understandable that not every aspect of emerging targets or agents is included in this review. Nonetheless, there are several interesting studies to be added that may trigger broad interests in MM community:
5.1) “The Sec61 translocon is a therapeutic vulnerability in multiple myeloma. EMBO Molecular Medicine, 2022”. While studying a neglected tropical disease, Buruli ulcer, researchers discovered a novel therapeutic target for multiple myeloma that could allow to bypass the resistance of proteasomal inhibitor.
5.2) “Proteomic profiling reveals CDK6 upregulation as a targetable resistance mechanism for lenalidomide in multiple myeloma. Nat Commun 2022”. This work identifies CDK6 upregulation as a druggable target in IMiD-resistant multiple myeloma and highlights the use of proteomic studies to uncover non-genetic resistance mechanisms in cancer.
6. Other points easy to fix:
6.1) Citations are better to be put before the period.
6.2) Several misspelling BCMA as BMCA.
6.3) Figure 3. The authors used a female figure to illustrate the immunotherapies in MM while MM could be found in any sex group. Recommend changing it to a unisex figure.
In summary, this manuscript is well written and adds great value to the current basic research and clinical practice of MM and is recommended for publication on Cancers after the revisions mentioned above.
Author Response
Reviewer 4:
Some concerns or comments for the current version:
- The authors did not mention the traditional chemotherapies approved for MM, although this is not the focus of the review according to the title. However, as a comprehensive review, the manuscript should at least contain a short paragraph or a table illustrating what chemotherapies have been used for MM patients, such as alkylating agent.
Answer: Added a new section about alkylating agents, Page 5, lines 180-201.
- In the section of immunotherapy, it is worthy to introduce the peptide vaccines (including but not limited to, ImMucin, PVX-410, and PD-L1 peptide IO103) that had been tested in clinical trials of MM patients.
Answer: Please see new “peptide vaccines” section starting at page 14, line 609.
- In the section of “5. Targeted Therapies and Small Molecules”, the authors elaborated different categories of targeted inhibitors by introducing the mechanisms and clinical significance. One thing to add is the emerging technique of PROTAC (proteolysis-targeting chimera), which represents a novel category of small molecule to degrade the targets, instead of merely inhibiting the targets. There are several review or research articles discussing the applications of PROTAC molecules in MM treatment. This is a non-neglectable trend in MM therapy.
Answer: Please see new “Proteolysis-targeting Chimera” section starting at page 19, line 767.
- Regarding the heterogeneity of MM cells, scientists have already applied single cell sequencing to identify resistance pathways and therapeutic targets (e.g. Nat Med 2021, 27: 491-503). The authors mentioned “using the power of next generation sequencing coupled with genome editing” in conclusion remarks, but it could be better if they add one section before the conclusion to elaborate one or two studies.
Answer: please see additions: (1) page 8, line 107-134, (2) page 20, line 825-838.
- The MM research community are keeping proposing potential targets, so it is understandable that not every aspect of emerging targets or agents is included in this review. Nonetheless, there are several interesting studies to be added that may trigger broad interests in MM community:
5.1) “The Sec61 translocon is a therapeutic vulnerability in multiple myeloma. EMBO Molecular Medicine, 2022”. While studying a neglected tropical disease, Buruli ulcer, researchers discovered a novel therapeutic target for multiple myeloma that could allow to bypass the resistance of proteasomal inhibitor.
Answer: please see addition – page 22, line 845-859.
5.2) “Proteomic profiling reveals CDK6 upregulation as a targetable resistance mechanism for lenalidomide in multiple myeloma. Nat Commun 2022”. This work identifies CDK6 upregulation as a druggable target in IMiD-resistant multiple myeloma and highlights the use of proteomic studies to uncover non-genetic resistance mechanisms in cancer.
Answer: please see addition – page 23, line 862-870.
Round 2
Reviewer 2 Report
After revision, the quality of this manuscript was significantly improved, and can reach the required quality standard of Cancers in my opinion. I recommend accepting without further revisions.